# Applying Data Mining Techniques for Predicting Prognosis in Patients with Rheumatoid Arthritis

**DOI:** 10.3390/healthcare8020085

**Published:** 2020-04-03

**Authors:** Chien-Ting Wu, Chia-Lun Lo, Chien-Hsueh Tung, Hsiu-Lan Cheng

**Affiliations:** 1Department of Pharmacy, Dalin Tzu Chi Hospital, Buddhist Tzu Chi Medical Foundation, Dailin, Chia- Yi 622, Taiwan; xelox78123@gmail.com; 2Institute of Healthcare Information Management, National Chung Cheng University, Minhsiung, Chia-Yi 621, Taiwan; 3Department of Health-Business Administration, Fooyin University, Kaohsiung City 83102, Taiwan; 4Department of Allergy, Immunology and Rheumatology, Dalin Tzu Chi Hospital, Buddhist Tzu Chi Medical Foundation, Dailin, Chia- Yi 622, Taiwan; dr5188@yahoo.com.tw; 5Department of Respiratory Therapy, Chang Gung University of Science and Technology, Puzi City, Chia-Yi 613, Taiwan; hlcheng@mail.cgust.edu.tw

**Keywords:** rheumatoid arthritis (RA), data mining techniques, decision tree, support vector machines, logic regression

## Abstract

Rheumatoid arthritis (RA) is a systematic chronic inflammatory disease. The disease mechanism remains unclear and may have resulted from autoimmune problems caused by genetic predisposing and pathogen infection. In clinical practice, selection of the initial treatment is based on the degree of disease activity, and treatment plans will be added gradually according to increased severity of the disease. However, treatment results can be unclear and treatment process uncertain and ambiguous, which can cause healthcare quality to become worse. This study attempts to combine expert opinions to construct various classifiers using a number of data mining techniques to analyze the different prognosis of two patient groups, by predicting whether the inflammatory indicator erythrocyte sedimentation rates of these two groups will be within the normal range with different medication strategies. Clinical data were collected for construction of different classifiers and we evaluate the prediction accuracy rate of each classifier afterwards. The optimum prediction model is selected from these classifiers to predict the prognosis of RA within these treatment strategies and analyze various results. The results show the accuracy rate of the prediction model by Logistic, SVM and DT module were 0.7927, 07829 and 0.9094, respectively. In the RA complications dataset, the accuracy rate of were 0.9393, 0.9290 and 0.9812, respectively. Futhermore, gain ratio was used to further analyze the rules and to discover which branch nodes are the most importance factor. The results of this study are helpful for formulation and development of guidelines for clinical RA treatments, and implementation of a decision support system by using the prediction model can assist medical staff to make correct decisions in the disease’s early stage.

## 1. Introduction

Rheumatoid arthritis (RA) is a systematic chronic inflammatory disease. The disease mechanism remains unclear and may have resulted from autoimmune problems caused by genetic predisposing and pathogen infection. It is characterized by joint swelling, joint tenderness and destruction of synovial joints, leading to severe joint deformity [1]. If patients have 1 to 4 of 7 criteria present for more than 6 weeks, the patient would be determined to have RA. The seven criteria are as below: (1) Morning stiffness of the joints and discomfort around the joint areas for more than an hour; (2) Arthritis in at least three or more joint areas; (3) Arthritis in hand joints; (4) Symmetric arthritis; (5) Rheumatoid nodules; (6) A positive serum rheumatoid factor (RF) test result; (7) Typical changes in hand or wrist joint radiographs [2]. 

The ACR treatment recommendation standard provides an important platform for guidelines on the process of making treatment decisions for RA. The standard also offers general treatment guidelines for initial medication, replacement or addition of medications for RA patients [3]. Drugs used for RA can be divided into non-steroidal anti-inflammatory drugs (NSAIDs), steroids, disease-modifying antirheumatic drug (DMARDs) and biologics. DMARDs are the major drugs used to treat RA, among which methotrexate is the most important drug and is used as the second-line drug when NSAIDs therapy is ineffective (because methotrexate acts quickly, starts working within a few weeks and has significant effects). According to the general treatment guidelines, steroids will be added when the usual treatment with DMARDs is ineffective during initial therapy because the full effect of DMARDs may not occur until after a certain period of time (around 2 to 3 months). Therefore, low dose steroids (≦7.5 mg) are used as bridge therapy to control disease progression until DMARDs are fully effective and then are gradually reduced once DMARDs start working. Most DMARDs can be used alone for monotherapy or be combined with two or more other DMARDs for multidrug therapy, and results indicate that multidrug therapy significantly reduces disease progression, particularly disease activity variables such as radiographic progression [3,4,5]. Biologics will be considered if disease remission can’t be achieved after combined DMARDs treatment for 3-6 months. In short, DMARDs are the most commonly used drugs for treating RA, while NSAIDs and steroids are auxiliary drugs to alleviate the symptoms of RA such as pain and inflammation.

In recent years, physicians have aggressively prescribed DMARDs to treat early-stage RA patients in order to delay the rate of articular cartilage and bone destruction. However, the autoimmune mechanism of RA was not targeted and treated. Moreover, the immune suppression effects of DMARDs are systematic rather than specific. Consequently, this treatment results in numerous side effects. The gradually developed biologics have more targeted effects and fewer side effects and thus medical experts are now searching for a more advanced treatment plan to customize the care plan and predict the prognosis of the disease by improving the underlying autoimmune response induced by RA [5,6].

This study included these four categories of RA drugs as variables and aimed to integrate blood biochemical tests for RF-IgM and ESR, and other variables (biochemical testing results such as GPT) and to cover all RA drugs (NSAIDs, steroids, DMARDs and biologics). In addition, we used data mining techniques to analyze RA patients with complications and without complications (cardiovascular diseases, hepatitis and ESRD) and predict whether the inflammatory indicator ESR values of these two groups of patients will be normal after treating with the four categories of RA drugs. The collected data was used to construct different classifiers which were then used to predict the accuracy rate, and the optimum prediction model was selected for prediction of RA prognosis and analysis of various results. 

This study organized the information relevant to the research topic by literature search (including RA diagnosis and prognosis) while collecting and processing the source of data. Next, in combination with expert opinions, we construct different classifiers using different data mining techniques and select the optimum prediction model based on the accuracy rates of these classifiers for prediction of RA prognosis and analysis of various results. The purpose of this study is to assist medical staff to make correct clinical decisions in the disease’s early stage by consideration of expert opinions, utilizing the data collected from a regional hospital in Taiwan’s Yunlin and Chiayi areas as the empirical basis, and various data mining techniques to predict the degree of inflammation and disease activity in patients without and with complications (cardiovascular diseases, hepatitis and ESRD) after treatment. The main research topics include: (1)Using data mining techniques to analyze patients without and with complications (cardiovascular diseases, hepatitis and ESRD) and to predict whether the ESR value will be within the normal range (refers to improvement of the degree of inflammation and corrosion of limbs).(2)Using data mining techniques and logical regression analysis to predict the prognosis of RA patients.

## 2. Rheumatoid Arthritis (RA) Associated Complications and Prognosis

RA results from autoimmune dysfunction caused by genetic predisposing and pathogen infection and may affect other body functions and lead to other complications. Futhermore, the complications caused by RA may influence clinical decisions about drug prescription and induce worse prognosis. 

Past studies had also utilized data mining methods for predicting the prognosis. For example, Mark, Deborah, and Alan (2005) used the decision tree to classify and evaluate the effect of the prognostic factor of ACR standards in 1987 [7]. Fautrel et al. (2009) also used the decision tree to explore the selection of the second-line drug DMARDs by rheumatologists based on the degree of disease severity after Methotrexate failed to treat RA patients [8]. Boyesen et al. (2009) analyzed the correlation among the blood test results of anti-CCP, RF, ESR, CRP and DXR- BMD, and utilized digital X-ray radiogrammetry (DXR) to predict the bone mineral density (BMD) of RA patients [9]. Weinblatt, Keystone, Cohen, and Freundlich (2011) examined the correlation between RF and radiographic progression of RA patients at week 52 after treatment with methotrexate for 12 weeks [10]. The 169 subjects were analyzed by Classification and Regression Tree (CART) after the 12-week treatment for determination of prognosis. Drouin and Drouin & Haraoui (2010) investigated the clinical outcome and predicted the radiographic progression of RA adult patients treated with methotrexate single therapy and found that the addition of DMARDs to methotrexate therapy in advance predicted good clinical outcome, whereas early-stage RA patients with a positive RF results and who are smokers showed poor clinical outcome [11]. Combe et al. (2001) studied the effects of the prognostic factors of radiographic joint damage and radiographic progression in early-stage RA and the 191 subjects with short-term illness (less than a year) were monitored for three years [12]. The variable used in logical regression analysis were the points of three-year radiographic progression. The results indicated that prognostic factors can identify radiographic joint damage in RA patients in early stage and predict three-year radiographic joint damage and radiographic progression. 

As noted in the above study, using data mining techniques to build the prediction model for the prognosis of RA is very common. Nevertheless, to the best of our knowledge, there is no study to compare and identify RA patients with cardiovascular disease, hepatitis, and ESRD, and to analyze RA patient groups without and with those complications, and to predict whether the ESR value will be within the normal range after treatment with the four categories of drugs in different groups of RA patients.

## 3. Research Method

This study combined expert opinions to construct various classifiers based on the collected data by using data mining techniques to analyze whether the ESR value will be within the normal range in RA patients after treating with the four categories of drugs. RA patients from the outpatient department were included as the subjects by the researcher and the therapeutic benefits of drugs which are crucial for the prognosis of RA are taken into account. Later, different classifiers were constructed to compare which is the optimum prediction model by comparing accuracy, sensitivity and specificity.

### 3.1. Data

In this study, we collected the complete records of outpatients with RA (ICD-9-CM = 714.0) as our research sample in two medical centers in southern Taiwan from January 2002 to December 2010. Each clinical record of RA patients contains demographics, such as gender, age, ESR, GPT and SCr biochemical testing results from Laboratory Information System (LIS); medical history, such as cardiovascular disease, hepatitis and ESRD; and all prescribed RA drugs (NSAIDs, steroids, DMARDs and biologics).

#### 3.1.1. Data Cleanup and Sample Selection

Each attribute of this study was in accordance with the literature and each case was also verified by domain experts. However, the 45,310 collected data samples from the case hospital may have various errors, including incomplete or inconsistent data, typos or mistakes, resulting from different data source formats. In this data-preprocessing step, the entire data will be deleted when a missing value was found, to avoid erroneous results. In addition, outliers with significant differences and repeated attribute codes that usually resulted from typing errors by clinical users will also be removed. The sample selection procedure is shown in Figure 1. Finally, 3486 cases were included as our experimental samples in the study. 3081 cases were in the “RA without complications dataset” group, and 405 cases were in the “RA with the complications dataset” group respectively.

#### 3.1.2. Research Variables

The RA patients from the outpatient department of the Hospital (primary diagnosis ICD-9-CM of 714.0) are included as the research subjects. After obtaining the data, to observe the variations of ESR spectra, only medical records containing two or more ESR test results and in which the second ESR value is less than the first ESR were saved (also with RF-Ig M and GPT test results). Each record contains nine items for use as the measurable variables in this study, including patient’s age, gender, blood biochemical test results: RF-Ig M, ESR and GPT, and four categories of RA drugs: NSAIDs, steroids, DMARDs and biologics. Moreover, past medical history indicating whether the patient had complications (cardiovascular disease, hepatitis and ESRD) is also included. The data was presented in different patterns and the DMARDs section was further divided into five new sections: Methotreate, Sulfsalazine, Leflunomide, Hydroxychloroquine and Cyclosporine attribute. One more attribute of the Number section was added, and thus a total of five drug name attributes were demonstrated. 

Through discussion with the medical experts on setting the independent variables for this study, based on the purpose of this research, the study is designed to predict whether the clinical test ESR result will be within the normal range (<12 mm/hr) in RA patients without and with complications (cardiovascular disease, hepatitis and ESRD) (dependent variables) after treating with the four categories of drugs. Finally, the definitions of various variables and data patterns are summarized and shown in Table 1. 

### 3.2. Experimental Design

The framework of this study adopted the supervised learning modules of WEKA3.6.5 open source application as a tool to predict suspected RA drug-induced complications model. The experimental design of this study made use of machine learning techniques, including Logistic Regression Model (LGR), Support Vector Machine Model (SVM) and Desion tree (DT) to construct the prediction model. Three prediction models were used separately to construct different single classifiers and the prediction results of these three techniques were used for comparison of the accuracy rate of prediction. 

For parameter setting, the value of minNumObj of J48 was set between 2 and 30 from “RA without complications dataset” and “RA with complications dataset”. The experiments indicated that the best performances were obtained when the value of minNumObj was adjusted to 38, 26 and 3, respectively. 

To accurately examine the performance of each classifier, this study adopted the 10-fold cross-validation [13,14] for experiments, and the accuracy rate was used as the index for comparison of the three model results. The average was calculated based on the performance results and represents the performance of the classifiers. 

This study used the 10-fold cross-validation for evaluation of the model classification performance, and the average was calculated based on the test performance results and represents the classification performance of the actual classification model, which can then be used for selection of models with better classification performance. We hope to construct a model for prediction of RA prognosis by using data mining techniques, and to predict the accuracy rate, specificity and sensitivity of different classification techniques for assessment. 

### 3.3. The Investigated Classification Techniques

The decision tree (DT) is a model utilizing classification and induction methods to generate a tree-like model and the common types of decision trees include Chi-square automatic interaction detector, classification and regression trees and C5.0 [15]. DT is a very useful data mining model, and its advantage is that it can process complex data and is not affected by linear regression and interactions between independent variables. Moreover, it can present the paths of individual independent variables and dependent variables to determine the nodes. To classify the input data, each node of the DT is a predicate, and each predicate can determine whether the variable is greater than or equal to or less than a certain value based on the input data and therefore each node can classify the input data into several types. The Classification and Regression Tree (CART) is a tree data structure proposed by Breiman in 1984 and builds a binary tree starting from the root by repeated iterations until it comes to the termination condition. When analyzing the data, if the selected data variable belongs to the categorical data, it is called a classification tree. If the selected data variable belongs to the continuous pattern, it is called a regression tree. DT mainly includes the following two steps: (1) build a tree data structure to train samples and classify the samples into different subsets based on the selected test conditions until all samples can be placed in the same category of a subset and the tree data structure is completed; (2) tree data structure pruning: perform pruning procedure from the root of the tree according to the constructed standards until the pruning standards are satisfied, and the results produce rules which are used for predicting the category of new samples. Once constructed, DT can quickly classify the test data starting from the root node, find the result through the branch of the tree based on the criteria of the test data, and lead to another internal node (leaf node). Its category label binds to the leaf node and DT can be used to track the classification results of the unlabeled data and finally locate the leaf node [13,16,17]. DT algorithms also provide a method for description of the test conditions for attribute types and presentation of the results of various attribute types. 

The support vector machines (SVM) concept established by Vapnik(1995) was mainly based on the Vapnik Chervonenks Dimension and structural risk minimization (SRM) of statistical learning theory and can solve realistic problems such as small sample size, non-linear, high dimension, local and minimum points [18]. SVM mainly utilizes the input training data and through the mechanism of learning can construct supervised learning techniques for classification or forecast of the regression function, raises the input variables and output variables to high vector space, and divides the space into separating hyperplane to separate the data into two or more classes and maximize the distance between the two or more classes so as to achieve the best classification effect. Its major function is to process the classification problems encountered during data mining. 

In quantitative analysis, logistic regression (LGR) is the most common statistical method. The LGR model is a widely used statistical technique and is used to forecast the value of a binary or ordinal variable, describe the relationship between coefficient and reaction coefficient and predict the response factor. In clinical research, observation of the problems is usually categorical data while the dependent variables of LGR model are binary variables, e.g., survival or death, metastatic or un-metastatic [13,19,20]. 

## 4. Results

### 4.1. Descriptions of the Related Variables

The descriptive statistics for the RA without complication dataset (research subjects include only RA patients) and the RA complication dataset (research subjects include RA patients and patients with cardiovascular disease, hepatitis and ESRD) are shown in Table 2 and Table 3, respectively. 

### 4.2. Evaluation Results

The two datasets of this study are RA and RA-COM and data mining analysis is performed by using Simple Logistic, SMO and J48 single classifier which represent the Logistic regression model, support vector machine, and decision tree classifier modules in the Weka software, respectively. 

In the RA dataset, we analyze and predict whether the ESR values of the RA patients without complications will be normal after treating with the four categories of drugs, and the accuracy rate of Simple Logistic, SMO and J48 model was 0.7927, 07829 and 0.9094, respectively, while the sensitivity and specificity are 0.792 and 0.793, 0.783 and 0.783, and 0.908 and 0.911, respectively. In the RA-COM dataset, we analyze and predict whether the ESR values of the RA patients with complications (cardiovascular diseases, hepatitis and ESRD) will be normal after treating with the four categories of drugs, and the accuracy rate of Simple Logistic, SMO and J48 model was 0.9393, 0.9290 and 0.9812, respectively, while the sensitivity and specificity are 0.935 and 0.936, 0.907 and 0.920, and 0.984 and 0.978, respectively. The analysis results of the RA dataset are shown in Table 3.

By comparison of the accuracy rate of prediction for the classifier Simple Logistic, SMO and J48, this study selected the optimum prediction model to forecast RA prognosis. The two datasets RA and RA-COM were separately resampled (30 groups) and the results were recorded. From each dataset’s 30 resamples, we selected the optimum resample group, adjusted the sample size to minimum by using the J48 single classifier correlation rule, and analyzed the 30 resamples. 

Among all single classifiers, the accuracy rate of classification for SMO is slightly lower than Simple Logistic and the accuracy rate of classification for J48 is higher than Simple Logistic. In summary, the J48 classifier showed the highest accuracy rate of prediction and its sensitivity and specificity is also close to 0.91 and 0.92 (up to 1 is best), respectively, and thus J48 is the optimum prediction model. 

The analysis results of the RA-COM dataset are shown in Table 4. Among all single classifiers, the accuracy rate of classification for SMO is slightly lower than Simple Logistic and the accuracy rate of classification for J48 is higher than Simple Logistic and SMO. In summary, the J48 classifier showed the highest accuracy rate of prediction and its sensitivity is also close to 0.99 (up to 1 is best), respectively, and thus J48 is the optimum prediction model. The classification tree analysis of the dataset is shown in Table 4. 

In addition to the comparison of the prediction performance using different techniques, we further evaluated the importance of each input variable for the purpose of clinical practice. For the RA and RA-COM datasets, we calculated the score of each input variable based on the gain ratio value associated with the dependent variable in Weka software and then ranked all input variables. 

Table 5 lists the rankings of input variables for both RA and RA-COM groups. As shown, in RA dataset, the ranks of the RA attributes in order are as follows: Sex, day_d and RF_IgM, Steroids, IF_NSAIDs, Sulfsalazine and Methotreate, Cyclosporine and IF_Biologics_i, Hydroxychloroquine and Leflunomide, and Age. The effects of Number and GPT on reduction of ESR to <12 mm/hr had the lowest explanatory power. Therefore, our results show the value of utilizing this data to make adequate decisions when clinicians make diagnoses for RA patients. 

### 4.3. Discussion

Whether the prediction accuracy of ESR value will be within the normal range in RA patients after treating with the four categories of drugs has been a heated topic among the RA clinical pathway guidelines [9,10,11,12]. Statistical models are commonly utilized in related literature and predicable accuracy has been improved by a data mining-related technology approach. Our study shows that decision tree based techniques can be more efficient and accurate than other traditional and function based classifiers, and these decision-tree-based approaches could be widely adopted to assist clinicians in making better medication decisions. 

Several factors are found to be crucial in making diagnosis for RA patients. The higher explanatory power factors are selected for further analysis with reference to relevant literature and are described for both RA groups and RA-COM groups as follows:

#### 4.3.1. RA-Group

The details are described as follows:Sex: no significant difference was found between males and females. However, the ESR of males reduced to below 12 more easily than the ESR of females in this study. From the report published by McInnes et al., annual incidence of RA was 40/100,000 in women while the rate in men is about half of the rate in women [1]. Nonetheless, in this study, male patients were found to be better in controlling the disease which is different from past reports, and thus further research and analysis are required.Day_d: based on experts’ clinical interpretations, the longer the treatment, the better the treatment efficacy and the ESR may be easily reduced to below 12. According to Mark et al., under the general treatment guidelines, the full effect of DMARDs may not occur until a certain period of time (around two to three months) during the initial therapy [20], and thus the explanatory power of day_d is high, which is similar to the result of this study.RF-IgM: based on experts’ clinical interpretations, RA patients with an RF-IgM > 40 g/L had severe conditions and poorly controlled disease and the ESR may not be easily reduced to below 12. According to Combe et al., the positivity of rheumatoid factor RF-IgM could predict the destruction of the joints and the radiographic progression in early-stage RA [12]. If the odds ratio (OR) of the predictor variable RF-IgM correlation is high (*P* < 0.001) [3], the explanatory power of RF-IgM is high, which is similar to the result of this study.Steroids: based on experts’ clinical interpretations, patients with better controlled disease require less doses and the ESR may be easily reduced to below 12. From the results reported by Gestel et al., combination of low dose (≦7.5 mg) steroids auxiliary treatment may improve disease prognosis [4,5] and the explanatory power of steroids is high, which is similar to the result of this study.

#### 4.3.2. RA_COM Group

The details are described as follows: RF_IgM: based on experts’ clinical interpretations, RA patients with an RF_IgM > 40 g/L had severe conditions and poorly controlled disease and the ESR may not be easily reduced to below 12. According to Combe et al., the positivity of rheumatoid factor RF-IgM could predict the destruction of the joints and the radiographic progression in early-stage RA. If the odds ratio (OR) of the predictor variable RF-IgM correlation is high (*P* < 0.001) [12], the explanatory power of RF_IgM is high, which is similar to the result of this study.Methotreate: based on experts’ clinical interpretations, patients with complications had better controlled disease after taking Methotrexate, and the ESR is more easily reduced to below 12. From the results reported by Mark et al. [7], the immunosuppressant methotrexate acts quickly, starts working within a few weeks and has excellent efficacy. Most patients respond well to the methotrexate treatment and therefore the methotrexate is currently the most frequently used first-line drug in RA [20] and the explanation power of Methotrexat is high, which is similar to the result of this study.Leflunomide: based on experts’ clinical interpretations, Leflunomide is the second-line drug and only used to treated severe patients in later treatment course and thus these patients frequently had poorly controlled disease and the ESR cannot be easily reduced to below 12. From the results reported by Mark et al., Leflunomide is the second-line drug and is specifically for treating severe RA patients [20] and the explanation power of Leflunomide is high, which is similar to the result of this study.IF_NSAIDs: based on experts’ clinical interpretations, RA patients with acute symptoms and severe disease activity will be treated with these drugs and thus these patients had poorly controlled disease and the ESR cannot be easily reduced to below 12. According to the results published by Belton and Mark et al., NSAIDs are widely used for treating pain and inflammation, e.g., sore throat, fever, gout, sprains and joint pain inflammation, including RA and osteoarthritis [3,21] and the explanation power of IF_NSAIDs is high, which is similar to the result of this study.

## 5. Conclusions

### 5.1. Results and Suggestions

The inappropriateness of drug use is a common influence on patient safety. Prediction of the adequate usage of medicines and patient prognosis after treatment is extremely important for improving healthcare quality and may decrease severe morbidity and mortality [22].

This study makes use of the LGR (Simple Logistic), SVM (SMO) and DT(J48) single classifier to analyze the accuracy rates of the two datasets indicating that the accuracy rate of DT is higher than LGR, which also demonstrates that data mining techniques are superior to conventional statistical analysis for prediction of the prognosis of RA patients. In the future, this prediction model can be used as an information platform to assist medical staff to make correct clinical decisions in an early stage. The attribute correlation of DT classifier can be analyzed by using important branch nodes so as to predict the importance of each attribute of a dataset, which is helpful for the formulation and development of treatment guidelines. In the analysis of the RA dataset and the RA-COM dataset, during the training process of classification models, data mining techniques were used to analyze RA patients with and without complications.

### 5.2. Research Contribution

The contributions to this study are listed below. To the clinicians, who can utilize data mining techniques to predict RA prognosis so as to assist medical staff to make correct clinical decisions in the early stage. Combining expert opinion and data mining techniques to predict the prognosis of RA patients can help the clinical care guideline maker to make modifications to the treatment strategy accordingly based on disease conditions, to better improve patients’ prognosis and extend patients survival time. 

### 5.3. Research Restrictions

This study is a retrospective from the RA database in a regional teaching hospital in Taiwan. The possibility of other factors affecting the RA treatment have not been considered. Therefore, the restrictions encountered during the studying process are described herein. 

First of all, clinically, RA results from autoimmune dysfunction and may affect other body functions which lead to other complications. However, RA complications are rather cumbersome and can be divided into inherited and acquired complications. Inherited complications are those that occurred before the confirmed diagnosis of RA by physicians and acquired complications are induced by autoimmune problems or drugs and therefore are not assessed in this study. Besides, there are several major illnesses and injury-associated complications not included in this study due to privacy issues in the case hospital.

### 5.4. Future Research Direction 

For comparison of the classification methods, the present study used all variables and the variables were selected as the factors for constructing our classification methods. However, we did not adjust the parameter values to adjust the evaluation results. Future studies may explore more of these aspects. 

Currently, whether the RA complications are inherited or acquired remains unidentified. Some studies mention that air pollution (PM 2.5) is also one of the factors influencing deterioration with rheumatoid arthritis over the years. Therefore, factors within the lifestyle of RA patients can also be considered one of the variables of the prediction model in future study.

## Figures and Tables

**Figure 1 healthcare-08-00085-f001:**
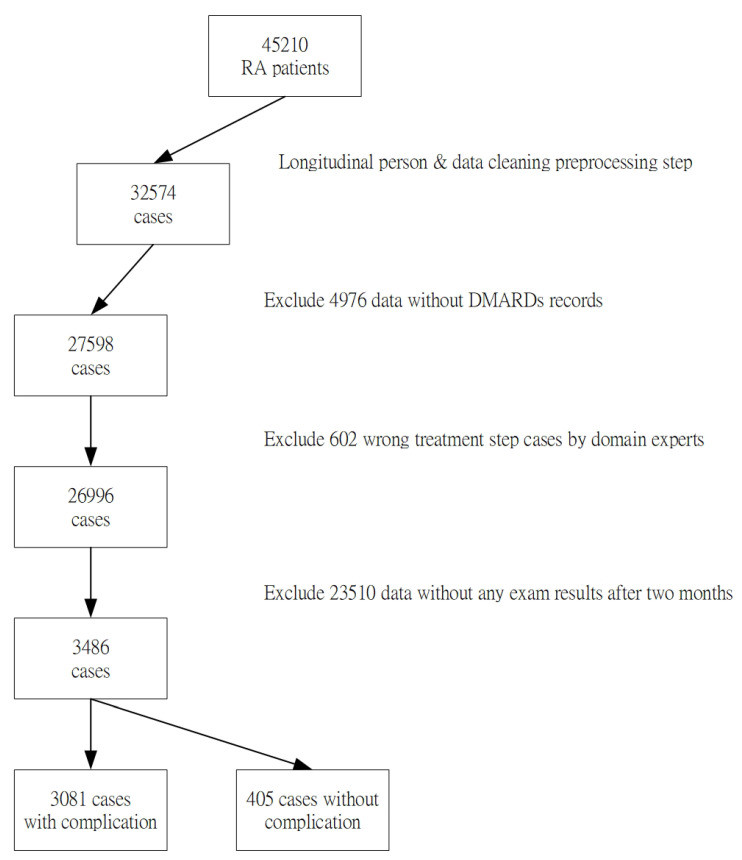
Flowchart showing derivation of the study sample.

**Table 1 healthcare-08-00085-t001:** Definitions of the variables used in this study.

Column Name	Column Description	Description of Definitions
Age	9 to 88	The date of diagnosis of RA minus the patient’s date of birth.
Gender	Male or female	Male: M Female: F
RF-Ig M	Rheumatoid Arthritis Index	The blood test results obtained before receiving treatments.
GPT	Liver function index	The blood test results obtained before receiving treatments.
IF_NSAIDs	The patient uses non-steroid classified into Cox-2 (Etodolac, Meloxicam, Nimesulide, Celecoxib, and Etoricoxib) and Non- Cox-2 (Aceclofenac, Diclofenac, Flubiprofen, Naproxen, and Sulindac) drugs or not.	If non-steroid is used record as yes, 0therwise, record as N.
Steroids	Steroids (include prednisolone and methylprednisolone).	Calculate the average doses of the prescribed steroids by examining the patient’s medical record between two ESR test dates of the current treatment.
DMARDs	The patient uses DMARDs (include Methotreate, Leflunomide, Hydroxychloroquine, Sulfsalazine and Cyclosporine) or not.	If a DMARD (divide five attributes: Methotreate, Sulfsalazine, Leflunomide, Hydroxychloroquine and Cyclosporine) is used, record as Y, otherwise record as N.
Number	The drugs should be the five drugs described in the above DMARDs section.	Calculate the total number of DMARDs the patient uses.
IF_Biologics_i	The patient uses biologics (including Etanercept, Adalimumab and Rituximab) or not.	If a biologic by examining the patient’s medical record between two ESR test dates of the current treatment is used, record as Y, otherwise record as N.
day_d	Duration of medication.	Calculate the days between two ESR test dates.
Cardiovascular diseases	Patient has cardiovascular disease or not.	ICD-9-CM: 401.9, 402.9, 405.99, 410.90, 412.0, 414.00, 414.9, 424.90, 428.0, 428.9, 429.2, 434.9, 434.91, 435.9, 436.0, 437.0, 437.2, 437.9, 438.9, 440.9.
Hepatitis	The patient has hepatitis or not.	ICD-9-CM: 070.30, 070.51, 070.9, 070.9.
ESRD	The patient has end-stage renal disease or not.	Secondary diagnosis ICD-9-CM: 585.6.
ESR	Is patient’s ESR normal?	The improved (reduced) ESR values obtained by examining the patient’s medical record between two ESR test dates of the current treatment. (normal range within 0–12 mm/h)

**Table 2 healthcare-08-00085-t002:** Descriptive statistics of rheumatoid arthritis (RA) without and with complication dataset.

Research Variables		RA Patients Without Complications n = 3081	RA Patients with Complications n = 405
Age (years)		57.18 (9 to 88) [8.78]	64.25 (41 to 86) [8.78]
Sex		M:475	M:60
		F:2,606	F:345
RF_IgM (g/L)		176.15 (0.57~3800) [300.33]	188.71 (4.23~2860) [419.15]
GPT (U/L)		24.66 (2~557) [24.15]	30.14 (0~40) [27.60]
IF_NSAIDs		Y:2,023	Y:342
		N:1,058	N:63
Steroids (mg)		5.72 (0~66) [6.01]	3.70 (0~16.5) [3.21]
DMARDs	Methotreate	Y:2635	Y:313
		N:446	N:92
	Sulfsalazine	Y:776	Y:310
		N:2,305	N:95
	Leflunomide	Y:654	Y:97
		N:2,427	N:308
	Hydroxychloroquine	Y:968	Y:123
		N:2,113	N:282
	Cyclosporine	Y:174	Y:16
		N:2,907	N:389
Number	Combination of 1 to 3 drug names	1:1,748	1:50
		2:540	2:256
		3:793	3:99
IF_Biologics_i (Biologics)		Y:59	Y:24
		N:3,022	N:381
day_d (Medication time) (day)		125 (60~2713) [150.20]	114 (60~1321) [88.37]
ESR < 12 mm/h		Y:845	Y:113
		N:2,236	N:292

Symbol Note—Age, RF_IgM, GPT, Steroids, day_d: μ [σ].

**Table 3 healthcare-08-00085-t003:** Comparison of the prediction results of different model for RA.

Single Classifier	Correctly Classified Instances		Sensitivity		Specificity	
	μ (σ)	Max/Min	μ (σ)	Max/Min	μ (σ)	Max/Min
Simple Logistic	0.7927 (0.019)	0.8198/0.7604	0.792 (0.059)	0.874/0.706	0.793 (0.054)	0.866/0.705
SMO	0.7829 (0.018)	0.8162/0.7500	0.783 (0.076)	0.901/0.679	0.783 (0.069)	0.878/0.667
J48	0.9094 (0.042)	0.9519/0.8098	0.908 (0.047)	0.959/0.785	0.911 (0.039)	0.961/0.829

**Table 4 healthcare-08-00085-t004:** Comparison of the prediction results of different model for RA-COM.

Single Classifier	Correctly Classified Instances		Sensitivity		Specificity	
	μ (σ)	Max/Min	μ (σ)	Max/Min	μ (σ)	Max/Min
Simple Logistic	0.9393 (0.068)	1.000/0.800	0.935 (0.090)	1.000/0.667	0.936 (0.083)	1.000/0.743
SMO	0.9290 (0.072)	1.000/0.7753	0.907 (0.109)	1.000/0.602	0.920 (0.087)	1.000/0.755
J48	0.9812 (0.024)	1.000/0.9012	0.984 (0.024)	1.000/0.892	0.978 (0.025)	1.000/0.909

**Table 5 healthcare-08-00085-t005:** Ranking of variables for the RA and RA-COM groups.

Variable	RA Group	RA-COM Group
GPT	13	13
Number	14	12
Age	12	10
Leflunomide	10	3
Hydroxychloroquine	11	8
Cyclosporine	7	7
If_Biologics_i	6	5
Methotreate	8	2
Sulfsalazine	9	9
IF_NSAIDs	5	4
Steroids	4	14
RF_igM	3	1
Day_d(Medication Time)	2	11
Sex	1	6

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
