# Peer review of "Applying Data Mining Techniques for Predicting Prognosis in Patients with Rheumatoid Arthritis"

_healthcare, 2020, doi:10.3390/healthcare8020085_

Round 1
Reviewer 1 Report
This is an interesting research paper dealing with Rheumatoid Arthritis.
However there are some major points to be studied by the authors:
- The text is difficult to read because there are too much abbreviations
- Rheumatoid Arthritis has also been discussed in the context of air pollution (PM 2.5). Nothing is told about that.
- Section 3.1 'Data' is not clear. How many Rheumatoid Arthritis patients are now involved?
- Section 3.1 'Data' Why are the data chosen from 2002-2010? What about a control with data from 2010-2018?
- Section 4.3 first paragraph is not clear what the message really is?
- Figure 1 is not described how the value of 14 is reached?
- Section 5.1 last sentence 'resulted from autoimmune problems or drugs is unclear and therefore future'. That's a strange finding and makes the whole study questionable
Author Response
We appreciate the referees’ suggestions and their time to help improve this paper. We have carefully revised this manuscript based on their comments. Below please find our item-by-item response to the referees’ comments. Thank you.
1 |
C: The text is difficult to read because there are too much abbreviations
|
R: Thanks for the referee’s comment. We have tried our best to modify the several paragraphs of the abstract and the whole manuscript. The modified sentences were highlight by yellow color to show their difference. |
|
2 |
C: Rheumatoid Arthritis has also been discussed in the context of air pollution (PM 2.5). Nothing is told about that. |
R: Thanks for the referee’s comment. Yes, Some evidence of literature to support air pollution (PM2.5) would be getting deterioration with rheumatoid arthritis. Nevertheless, in this study, the application scenario just focuses and evaluates which is the best personal treatment strategy and suitable for the different RA patients when clinicians make assessment. Besides, there are not any air pollution-related information that can be collected in the EMR system until now in any hospital of Taiwan. Therefore, air pollution (PM2.5) would be not included to be a factor in the evaluation model of this study. but we have added that information in the future study section. |
|
3 |
C: Section 3.1 'Data' is not clear. How many Rheumatoid Arthritis patients are now involved? |
R: Thanks for the referee’ comment, we had added sentences and figure 1 to show how much data we were collected and out data preprocessing steps which are shown in figure 1. We hope that figure can help the reader to know how much effort we did in this step.
“Each attribute of this study was in accordance with literature and each case was also verified by domain experts. However, the 45310 collected data from the case hospital may have various errors, including incomplete or inconsistent data, typo or the mistake resulting in different data source formats. In this data-preprocessing step, the entire data will be deleted when the missing value was founding to avoid erroneous results. In addition, outliners with significant differences, repeated attribute codes that usually resulted from typing errors by clinical users will also be removed. The sample selection procedure was shown in Figure 1. Finally, 3486 cases were included as our experimental samples in the study. 3,081 cases were under the RA without complications dataset" group, and 405 cases as "RA with the complications dataset" group respectively.” |
|
4 |
C: Section 3.1 'Data' Why are the data chosen from 2002-2010? What about a control with data from 2010-2018? |
R: Thanks for the referee’s comment. Between 2010 and 2016, it was the period of code conversion from ICD9 to ICD10 each hospital in Taiwan. Since the two coding systems cannot correspond one by one consistently, and there are plenty of mistakes in those periods, we cannot analyze the whole data set from 2002~2020 together. This is the reason why we just collected and analyze the patient sample from 2002~2010. |
|
5 |
C: Section 4.3 first paragraph is not clear what the message really is? |
R: Thanks for the referee’s comment. Yes, the paragraph addresses part of the evaluation result, we agree that the paragraph is not suitable discussion section (4.3). Therefore, we had been moved the whole paragraph into the section 4.2. |
|
6 |
C:. Figure 1 is not described how the value of 14 is reached? |
R: Thanks for the referee’s comment. But there is some misunderstanding for that because the 14 is just the value of the important ranking list of the factors. After we thorough deliberation, both of the information was not appropriate to present by the bar chart. We had adjusted and present by Table. |
|
7 |
C: Section 5.1 last sentence 'resulted from autoimmune problems or drugs is unclear and therefore future'. That's a strange finding and makes the whole study questionable |
R: Thanks for the referee’s comment. That is our mistake to leave the Incorrect sentences on there. We had been removed the superfluous sentences. |
Finally, thank you again for your insightful comments and help, which significantly improve the readability of this manuscript.
Reviewer 2 Report
This is a good study,
Do authors mean that first line therapy is methotrexate? Is that so?
Author Response
We appreciate the referees’ suggestions and their time to help improve this paper. We have carefully revised this manuscript based on their comments. Below please find our item-by-item response to the referees’ comments. Thank you.
1 |
C: Do authors mean that first line therapy is methotrexate? Is that so? |
R: Thanks for the referee’s comment. Yes, Methotrexate is first-line therapy medicine in the case hospital. However, Because of the drug properties of the methotrexate is heavier than others, it often relies on other medicine to be the "bridge" in clinical treatment guidelines. Therefore, Methotrexate will always be singly used after the DMRDs combine NSAIDs treatment failed. To sum up, even though Methotrexate is the first-line medication, but it is still kind of main considered medication when the second line choosing. |
Finally, thank you again for your insightful comments and help, which significantly improve the readability of this manuscript.
Round 2
Reviewer 1 Report
The authors answered the points raised by the reviewer and revised the paper accordingly.
Reviewer 2 Report
This paper is so far so good.
However this paper is so complicated.
I would prefer rather than simple papers.
Could you clarified cinical questions simple?